# Recent phenological shifts of migratory birds at a Mediterranean spring stopover site: Species wintering in the Sahel advance passage more than tropical winterers

Ivan Maggini[1]*, Massimiliano Cardinale[2], Jonas Hentati Sundberg[2], Fernando Spina[3], Leonida Fusani[1,4]

1 Austrian Ornithological Centre, Konrad-Lorenz Institute of Ethology, University of Veterinary Medicine Vienna, Wien, Austria, 2 Department of Aquatic Resources, Institute of Marine Research, Swedish University of Agricultural Sciences, Lysekil, Sweden, 3 Institute for Environmental Protection and Research (ISPRA), Ozzano dell'Emilia (BO), Italy, 4 Department of Behavioural and Cognitive Biology, University of Vienna, Wien, Austria

* ivan.maggini@vetmeduni.ac.at

**Data Availability Statement:** https://phaidra.vetmeduni.ac.at/view/o:114.

## Abstract

Spring migration phenology is shifting towards earlier dates as a response to climate change in many bird species. However, the patterns of change might not be the same for all species, populations, sex and age classes. In particular, patterns of change could differ between species with different ecology. We analyzed 18 years of standardized bird capture data at a spring stopover site on the island of Ponza, Italy, to determine species-specific rates of phenological change for 30 species following the crossing of the Mediterranean Sea. The advancement of spring passage was more pronounced in species wintering in Northern Africa (i.e. short-distance migrants) and in the Sahel zone. Only males from species wintering further South in the forests of central Africa advanced their passage, with no effect on the overall peak date of passage of the species. The migration window on Ponza broadened in many species, suggesting that early migrants within a species are advancing their migration more than late migrants. These data suggest that the cues available to the birds to adjust departure might be changing at different rates depending on wintering location and habitat, or that early migrants of different species might be responding differently to changing conditions along the route. However, more data on departure time from the wintering areas are required to understand the mechanisms underlying such phenological changes.

## Introduction

Migration phenology in birds and other animals has been shifting in recent years, along with overall climate change [1–4]. This is a global phenomenon observed in all continents where enough long-term data are available [5–9]. In the Palaearctic-African bird migration system, most studies documented an advance in spring migration and arrival at the breeding grounds

**Funding:** Funding across the study period was provided to LF by The Research Council of Norway grant N. 196451/V40 (https://www. forskningsradet.no/en/); the University of Ferrara, FAR 2008-2013 (http://www.unife.it/international); the Max Planck Society (https://www.mpg.de/en); and start-up funds of the University of Veterinary Medicine, Vienna (https://www.vetmeduni.ac.at/en/), and the University of Vienna (https://www.univie.ac.at/en/). The funders had no role in study design, data collection and analysis, decision to publish, or preparation of the manuscript.

**Competing interests:** The authors have declared that no competing interests exist.

[10, 11] though changes in autumn migration have been observed as well [12, 13]. Both short- and long-distance migrants are affected [14–16] and changes are usually related to changes in the North Atlantic Oscillation Index (NAOI) and temperatures along the route [14, 17–27]. The pattern of phenology shift is complex, and species, populations, sexes and age classes are affected differently [28–36]. Timing effects might be more pronounced in certain areas than in others [32, 33, 37, 38], and in some cases they might have opposite trajectories [39], possibly underlining weather effects along the route [40]. In some cases, the early phases of migration are affected more strongly than the late ones [23, 39, 41, 42]. It is still debated whether phenology shifts are driven by microevolutionary changes or by phenotypic plasticity [10, 11, 43, 44], though the latter mechanism has recently received increasing support [45].

Some of the methods used for detecting switches in phenology have been object of debate [46, 47]. Studies using first arrival dates might overestimate changes, and several authors advised to use median and percentile passage dates to better describe the phenomenon [16, 48, 49]. The latter approach provides tools to understand and monitor in more detail the process of phenology shifts in spring migration, which is likely linked to climate change. Conditions in the African wintering grounds are changing, e.g. the Sahel zone is becoming greener (as predicted by [50, 51], and described in [52]), while stopover areas in Northern Africa are becoming drier ([53–55], but see [56]). This might be due to the recent trend towards a positive NAOI in the last years (https://www.ncdc.noaa.gov/teleconnections/nao/, last accessed on July 2$^{nd}$, 2020). These environmental data suggest that species wintering in the Sahel and actively using stopover sites in Northern Africa might be more affected than others in their timing of passage, which should be reflected in an earlier arrival in Southern Europe.

Here, we aimed at identifying recent changes in migration phenology of migrants that cross the Mediterranean Sea, with a particular focus on within-species comparison between early and late migrants and on differences between species with different wintering areas. To this aim, we analyzed a large dataset of captures of migratory birds (nearly 220 000 individuals, mostly passerines) on spring migration from a small Italian island, where large numbers of individuals of several species are stopping over after crossing the Mediterranean Sea [57]. We calculated peak passage date and the dates of start and end of the main migration period for every year of the study, totaling 18 years, in 30 species of bird migrants. We determined the trends of change in these parameters for every species and tested for general patterns within groups of species based on their wintering range.

## Study site and methods

### Study site and ringing operations

This study was conducted on Ponza, a small island in the Tyrrhenian Sea (9.87 km$^2$) located about 50 km off Italy (40˚55' N, 12˚58' E), where spring bird migration has been monitored since 2002 (www.inanellamentoponza.it). Ponza attracts large numbers of African-European migratory landbirds during spring migration as it is located along one of the main Mediterranean migratory routes, with daily peaks of over 1500 individual birds ringed occurring several times during the study period. Birds were caught using mist-nets from March (or April in some years) to May (exact start and end dates are shown in S1 Table). Ringing was conducted under permit from the Regione Lazio (Determinazione Dirigenziale B0332/06; B0084/09; A12042/11; G00575/15; and G00668/18). No ethical permit is required for standard capture and ringing. Ringing was conducted daily except for days with heavy rain or strong winds (>15 knots). These conditions occurred on <1% of the total ringing days over the entire study period. The mist-nets were checked hourly from dawn until one hour after dusk. The average total length of mist nets deployed was 227 m. We kept the net brand (Lavorazione Reti

**Table 1. Summary of the study species captured on the island of Ponza between 2002 and 2019 and used in the analysis.**

| Species | Total individuals captured | Average individuals per year [min, max] | Median passage (Julian day) | Wintering area |
|---|---|---|---|---|
| *Acrocephalus arundinaceus* | 410 | 23 [4, 58] | 124.2 | Tropical |
| *Acrocephalus schoenobaenus* | 1162 | 65 [17, 236] | 127.1 | Sahel |
| *Anthus trivialis* | 2193 | 122 [19, 202] | 108.4 | Tropical |
| *Erithacus rubecula* | 13044 | 725 [1, 2599] | 86.5 | North Africa |
| *Ficedula albicollis* | 1548 | 86 [1, 279] | 112.7 | Tropical |
| *Ficedula hypoleuca* | 10312 | 573 [65, 914] | 115.8 | Tropical |
| *Hippolais icterina* | 6608 | 367 [45, 583] | 109.4 | Tropical |
| *Hirundo rustica* | 507 | 28 [1, 67] | 105.3 | Tropical |
| *Jynx torquilla* | 676 | 38 [17, 72] | 121.0 | Sahel |
| *Lanius senator* | 1811 | 101 [11, 214] | 105.3 | Sahel |
| *Luscinia megarhynchos* | 736 | 41 [13, 114] | 120.7 | Tropical |
| *Merops apiaster* | 8654 | 481 [46, 1052] | 130.4 | Tropical |
| *Muscicapa striata* | 18871 | 1048 [216, 2922] | 134.3 | Sahel |
| *Oenanthe hispanica* | 127 | 7 [0, 25] | 105.7 | Sahel |
| *Oenanthe oenanthe* | 2097 | 117 [21, 212] | 105.7 | Sahel |
| *Oriolus oriolus* | 1404 | 78 [19, 182] | 123.4 | Tropical |
| *Phoenicurus ochruros* | 1972 | 110 [0, 397] | 83.1 | North Africa |
| *Phoenicurus phoenicurus* | 5474 | 304 [45, 726] | 110.4 | Sahel |
| *Phylloscopus collybita* | 3809 | 212 [1, 589] | 86.9 | North Africa |
| *Phylloscopus sibilatrix* | 14784 | 821 [159, 1430] | 116.4 | Tropical |
| *Phylloscopus trochilus* | 12673 | 704 [83, 1321] | 110.0 | Sahel |
| *Saxicola rubetra* | 16980 | 943 [455, 1647] | 119.5 | Tropical |
| *Saxicola torquatus* | 564 | 31 [0, 147] | 74.9 | North Africa |
| *Streptopelia turtur* | 564 | 31 [16, 49] | 123.0 | Sahel |
| *Sylvia atricapilla* | 2303 | 128 [4, 574] | 98.4 | North Africa |
| *Sylvia borin* | 49713 | 2762 [581, 5967] | 130.2 | Tropical |
| *Sylvia cantillans* | 8019 | 446 [29, 1089] | 97.2 | Sahel |
| *Sylvia communis* | 30496 | 1694 [500, 3594] | 121.3 | Sahel |
| *Turdus philomelos* | 1828 | 102 [0, 477] | 82.8 | North Africa |
| *Upupa epops* | 458 | 25 [1, 64] | 91.9 | Sahel |

Total number of birds captured, yearly average with minima and maxima in brackets, median passage date expressed as Julian date (1 = January 1$^{st}$), and main wintering area are indicated.

Bonardi, Monte Isola BS, Italy, http://www.vbonardi.it/) and model (2.4 m height, 16 mm mesh size) constant throughout the entire study period. The birds were ringed, aged and sexed according to the available literature [58, 59]. We analyzed 18 years (2002–2019) of capture data standardized by daily effort (Catch Per Unit of Effort, hereafter CPUE). For this analysis, we used data of the 30 most abundant species in number of individuals during the study period (Table 1). We divided the species in three groups based on their main wintering area, referring to [60, 61] for a description of their wintering areas. We divided them into species wintering mainly in North Africa (North of the Sahara Desert), the Sahel zone (dry scrubland just South of the Sahara Desert), and tropical Africa (Guinea savanna and tropical forests).

## Analysis of passage timing

To define the peak date of passage and the time window encompassing the main migration period (hereafter referred as migration window), we used a 7-days Moving Average (MA) of

the daily CPUE values for each species (see S1 Fig for a visual representation of the general patterns). For every year of the study and for each species separately, peak date of passage was defined as the day with the highest MA of migrating birds. The start and end of the main migration period were defined from the tail ends of the timing distribution as the dates when MA was below 10% of the peak value. In years when the ringing season started after the onset of the migration period (i.e. with smoothed data for the first day above 10% of peak MA) and/ or ended before the end of the migration period (i.e. with smoothed data for the last day above 10% of peak MA) for any species, the tails of the distributions, and therefore the start or the end of the migration period, were not defined for that year and species. The width of the migration window was defined as the difference in days between the start and the end of the main migration period. Statistical analysis for changes in timing were made based on the annual values of start, peak and end of the main migration period, when available. Species-years where the total number of captures was lower than 5 were excluded from the analysis. This method is insensitive to the shape of the probability distribution of daily migration values and is therefore preferred over methods that rely on a pre-defined probability distribution (e.g. a normal distribution), and allows to identify migration peaks without relying on quantile measurements such as the median, which is not reliable in case of a truncated sample. S1 Fig shows that there is a nearby perfect overlap between the observed data and the fitted data using the MA method.

We analyzed changes in timing of the annual values of start, peak and end of the main migration period separately for every species using linear regressions. The slope of this regression indicates the average yearly change in date of passage on Ponza. Negative slopes indicate an advance in passage, while positive slopes indicate a delay. We compared the changes in passage dates of the three wintering groups (North Africa, Sahel, and Tropical Africa) using linear mixed effects models (LMM) with respectively start, peak, end of migration period, and migration window as response variables, and wintering group, year, and the interaction between wintering group and year as fixed effects, while species was used as a random effect. A significant interaction indicates different slopes of passage date over the years in the different groups. To test for pairwise differences, we first estimated the marginal means of the linear trends between wintering group and year using the *emtrends* function of package *emmeans* in R [62]. We then used the *cld* function of package *multcomp* [63] to compare these means pairwise among groups. This function groups the different levels of a variable (in our case the wintering groups) according to a set p-level, which is 0.05 by default. To estimate the p-level of non-significant pairwise comparisons we therefore had to change the p-level for grouping, thus resulting in p-values that are within a range rather than being exact (see Results). For the comparison of the start of the migration period and the migration window, we excluded the species wintering in North Africa, since for most of them the date of start of passage was not estimated.

Sex could be determined based on morphology in 11 species (1 from North Africa, 4 from the Sahel, and 6 from Tropical Africa). In these species, in addition to the analysis described above, we performed separate linear regressions for each sex in every species. We then compared species wintering in the Sahel and in Tropical Africa (we excluded the only species from North Africa for this analysis) in the slopes of the peak date of passage separately for males and females, using LMMs with peak date of passage as a response variable, year, wintering group, and the interaction term of year and wintering group as fixed effects, and species as random effect. Again, a significant interaction term would imply a different slope in the change over the years, and we compared groups using the same procedure as described above. We also compared males and females within each wintering group in a similar fashion, this time using

sex, year, and the interaction of sex and year as fixed effects. All analyses were performed using R 4.0.0 (www.r-project.org).

## Results

The slopes of change in passage dates for each species are shown in Table 2 and visualized in S2 Fig. During the study period, peak date of passage was advanced significantly in 5 of the 30 species used in our analysis (3 from the North African wintering group, 2 from the Sahel group), with additional two species with an advance that was close to significance (both from the Sahel group). The range of yearly advance in peak date of passage (for significant trends only) was between 1.0 and 1.4 days per year.

The start of the main migration period was significantly advanced in 6 species (3 from the Sahel group, 3 from the Tropical group), and close to significance in one additional species (from the Tropical group). Note that the start of the migration period was not determined in any of the species in the North African group due to fact that their passage almost invariably began before the start of the capture season on Ponza. The range of yearly advance in the start of migration was between 0.7 and 1.7 days per year.

There was a significant advance of the end of the migration period by 1.6 days per year in one species from the Sahel group, and a significant delay in the end of migration by 0.5 days per year in one species from the Tropical group. One species from the Sahel group had a close to significant delay in the end of the migration period.

The migration window was significantly broadened in two species (one from the Sahel and one from the Tropical group), while the broadening of the migration window was close to significance in one additional species from the Tropical group. In one species from the Sahel group, the migration window was almost significantly narrower.

The marginal mean slopes of the peak date of passage change per year were significantly different from zero in two of the wintering groups (North Africa: -0.8 ± 0.2 days per year, $p < 0.001$; Sahel: -0.5 ± 0.1 days per year, $p < 0.001$), while this was not the case for the Tropical group (-0.1 ± 0.1 days per year, $p = 0.433$) (Fig 1). The slopes of the North Africa and the Tropical groups differed significantly from each other (*cld* comparison: $p < 0.05$), while the difference between the Sahel and the Tropical groups was marginally non-significant ($0.05 < p < 0.06$). The slope of the Sahel group and of the North Africa group did not differ from each other ($0.20 < p < 0.25$). The overall effect of year across species on the peak passage was significant in the LMM ($t = -4.283$).

Peak date of passage was advanced overall in males (LMM: $t = -3.081$, Fig 2) but the slopes did not differ between Sahel and Tropical wintering birds (LMM; $t = 1.149$). The marginal mean slope of peak date of passage for males of the Sahel group was significantly different from zero (-0.7 ± 0.2 days per year, $p = 0.002$) and marginally non-significant in males of the Tropical group (-0.3 ± 0.2 days per year, $p = 0.061$). In females, there was both an overall effect of year on peak date of passage (LMM: $t = -2.609$, Fig 2) and on the interaction between year and wintering group (LMM; $t = 2.382$). The marginal mean slope of peak date of passage for females of the Sahel group was significantly different from zero (-0.6 ± 0.2 days per year, $p = 0.010$) and non-significant in females of the Tropical group (0.1 ± 0.2 days per year, $p = 0.559$).

When comparing sexes within the Sahel wintering group, there was no difference in slope between males and females (LMM: $t = -0.253$), while the slopes were different between sexes in the Tropical wintering group (LMM; $t = -2.050$). At the species level, peak date of passage was advanced significantly in both males and females of two species in the Sahel wintering group (Table 3).

**Table 2. Summary of the changes (in days per year) of start, peak, end date of the main migration period and migration window for 30 bird species migrating through the island of Ponza between 2002 and 2019.**

| Species | Start [days/year] | Peak [days/year] | End [days/year] | Migration window [days/year] |
|---|---|---|---|---|
| **Wintering group: North Africa** | | | | |
| *Erithacus rubecula* | NA | **-1.0 ± 0.4**\* | -0.5 ± 0.4 | NA |
| *Phoenicurus ochruros* | NA | -0.4 ± 0.3 | -0.3 ± 0.4 | NA |
| *Phylloscopus collybita* | NA | **-1.3 ± 0.5**\* | -0.3 ± 0.6 | NA |
| *Saxicola torquatus* | NA | -0.4 ± 0.3 | 0.2 ± 0.3 | NA |
| *Sylvia atricapilla* | NA | -0.3 ± 0.4 | -0.3 ± 0.4 | NA |
| *Turdus philomelos* | NA | **-1.3 ± 0.6**\* | -0.1 ± 0.8 | NA |
| **Wintering group: Sahel** | | | | |
| *Acrocephalus schoenobaenus* | -0.2 ± 0.4 | 0.3 ± 0.3 | 0.0 ± 0.9 | -0.5 ± 1.3 |
| *Jynx torquilla* | **-1.5 ± 0.5**\* | -0.3 ± 0.5 | -0.4 ± 0.3 | 1.2 ± 1.0 |
| *Lanius senator* | 0.1 ± 0.3 | -0.2 ± 0.3 | 0.6 ± 0.4 | 0.5 ± 0.3 |
| *Muscicapa striata* | -0.2 ± 0.4 | 0.5 ± 0.3 | NA | NA |
| *Oenanthe hispanica* | -0.3 ± 0.4 | -1.2 ± 0.7 | **-1.6 ± 0.5**\* | -1.2 ± 0.7 |
| *Oenanthe oenanthe* | -0.1 ± 0.5 | -0.7 ± 0.6 | -0.2 ± 0.3 | -0.1 ± 0.6 |
| *Phoenicurus phoenicurus* | **-1.6 ± 0.5**\* | **-1.4 ± 0.5**\*\* | -0.1 ± 0.3 | 0.2 ± 0.8 |
| *Phylloscopus trochilus* | 0.0 ± 0.8 | -0.7 ± 0.4 | 0.3 ± 0.2 | 0.0 ± 1.0 |
| *Streptopelia turtur* | -0.3 ± 0.3 | -0.1 ± 0.3 | 0.2 ± 0.4 | 0.5 ± 0.8 |
| *Sylvia cantillans* | 0.6 ± 1.0 | **-1.0 ± 0.4**\* | -0.1 ± 0.5 | -2.4 ± 0.8 |
| *Sylvia communis* | -0.5 ± 0.3 | -0.1 ± 0.3 | -0.2 ± 0.7 | 0.7 ± 1.0 |
| *Upupa epops* | **-1.3 ± 0.5**\* | -1.1 ± 0.5 | 0.7 ± 0.6 | **3.9 ± 1.1**\* |
| **Wintering group: Tropical Africa** | | | | |
| *Acrocephalus arundinaceus* | **-0.7 ± 0.2**\*\* | -0.6 ± 0.4 | 0.1 ± 0.2 | 0.7 ± 0.4 |
| *Anthus trivialis* | -0.7 ± 0.6 | -0.5 ± 0.3 | -0.2 ± 0.3 | 0.8 ± 1.4 |
| *Ficedula albicollis* | -0.8 ± 0.5 | -0.4 ± 0.4 | -0.2 ± 0.3 | 0.5 ± 0.7 |
| *Ficedula hypoleuca* | **-0.7 ± 0.3**\* | 0.1 ± 0.3 | **0.5 ± 0.2**\* | 0.8 ± 0.4 |
| *Hippolais icterina* | 0.2 ± 0.2 | 0.4 ± 0.3 | 0.3 ± 0.7 | 0.2 ± 1.4 |
| *Hirundo rustica* | **-1.7 ± 0.4**\*\* | -0.5 ± 0.5 | 0.0 ± 0.3 | **2.6 ± 0.8**\* |
| *Luscinia megarhynchos* | -0.5 ± 0.4 | -0.4 ± 0.4 | 0.1 ± 0.3 | 0.7 ± 0.8 |
| *Merops apiaster* | -0.3 ± 0.2 | -0.1 ± 0.3 | 0.1 ± 0.3 | 0.4 ± 0.4 |
| *Oriolus oriolus* | -0.2 ± 0.2 | 0.4 ± 0.3 | NA | NA |
| *Phylloscopus sibilatrix* | -0.3 ± 0.4 | 0.3 ± 0.3 | 0.3 ± 0.2 | 1.2 ± 0.7 |
| *Saxicola rubetra* | -0.6 ± 0.3 | 0.1 ± 0.3 | 0.1 ± 0.1 | 0.6 ± 0.5 |
| *Sylvia borin* | -0.3 ± 0.3 | 0.2 ± 0.3 | NA | NA |

Slopes ± SE from the linear regression of date by year are given. Significant slopes are represented in bold typeface.

\*\* = p < 0.01

\* = 0.01 < p < 0.05. Exact p-values are shown in S2 Table. Negative values indicate an advanced passage. In the last column (Migration window), the magnitude of the change in width of the migration window is shown. Positive values indicate a broader migration window. NA indicates missing values when either start or end of the main migration period were not assessed.

Both the Sahel and the Tropical wintering groups significantly advanced the start of the migration period (Fig 1), so that there was an overall significant effect of year (LMM: t = -3.766), but the slope was not different between the two groups (LMM: t = -0.214). The marginal mean slopes were -0.5 ± 0.1 days per year in the Sahel group (p < 0.001) and -0.5 ± 0.1 days per year in the Tropical group (p < 0.001).

The marginal mean slope of the change in the end of the migration period was not significantly different from zero in any of the wintering groups (North Africa: -0.3 ± 0.2 days per

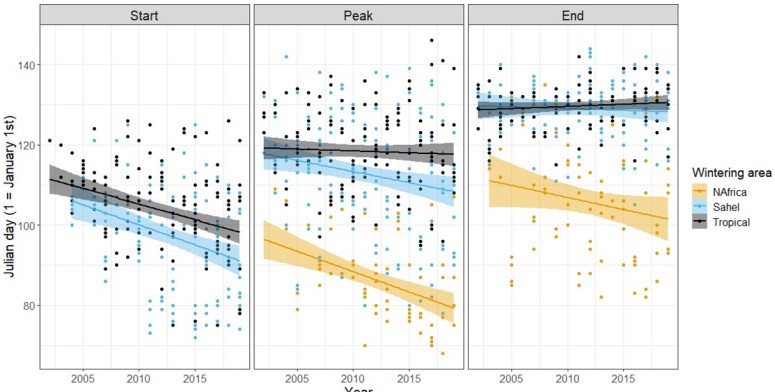

**Fig 1. Trends of change in start, peak, and end date of the main migration period of 30 bird species migrating through the island of Ponza between 2002 and 2019, subdivided into wintering groups.** Every dot represents the passage date of a single species in any given year. Patterns for all species singularly are shown in S1 Fig.

year, p = 0.074; Sahel: -0.1 ± 0.1 days per year, p = 0.678; Tropical: 0.1 ± 0.1 days per year, p = 0.373). There was no overall effect of year across species in the end of the migration period (LMM: t = -1.793) (Fig 1).

The migration window was broadened overall by 0.5 ± 0.3 days per year, though not significantly so (LMM: t = 1.818). There were no differences between wintering groups (LMM; t = -0.662) nor in the slopes of change between groups (LMM; t = 0.653).

## Discussion

After having been reported in a large number of studies in the early 2000's [10, 11], the advance of spring passage in migratory European-African migratory birds has received decreased attention, in particular in the Mediterranean basin. However, our results clearly show that this phenomenon is still ongoing, and it is occurring at a substantial rate. The values of yearly change in our study should be considered with caution for species with relatively low numbers of yearly captures or species for which data were not obtained every year. However, the robust overall results indicate that on average the peak of passage has been advanced by up to one day per year. The advance was most marked in species wintering in North Africa and,

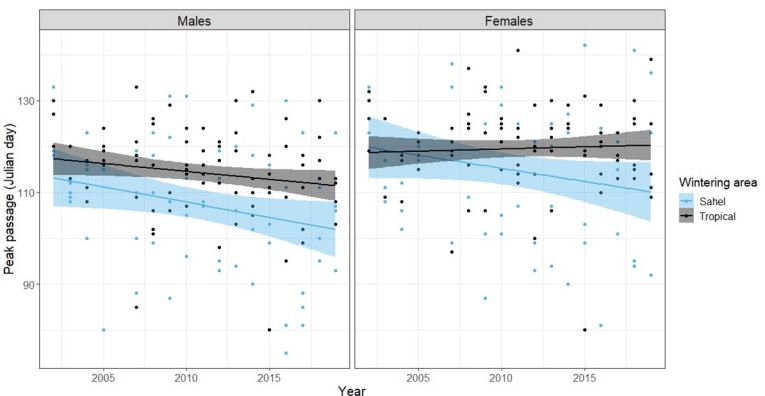

**Fig 2. Trends of change in the peak date of the main migration period on Ponza of males and females from 4 species wintering in the Sahel and 6 species wintering in tropical Africa.** Every dot represents the passage date of a single species in any given year.

**Table 3. Summary of the changes (in days per year) in the peak date of the main migration period of 11 sexually dimorphic migratory bird species on Ponza between 2002 and 2019, divided by sex.**

| Species | Males | | Females | |
|---|---|---|---|---|
| | Peak ± SE | p | Peak ± SE | p |
| **Wintering group: North Africa** | | | | |
| *Sylvia atricapilla* | -0.3 ± 0.5 | 0.618 | -0.6 ± 0.4 | 0.166 |
| **Wintering group: Sahel** | | | | |
| *Lanius senator* | -0.1 ± 0.4 | 0.856 | 0.9 ± 0.5 | 0.082 |
| *Oenanthe oenanthe* | 0.0 ± 0.6 | 1.000 | -0.5 ± 0.5 | 0.364 |
| *Phoenicurus phoenicurus* | **-1.6 ± 0.5** | **0.004**[**] | **-1.1 ± 0.5** | **0.046**[*] |
| *Sylvia cantillans* | **-1.0 ± 0.5** | **0.045**[*] | **-1.5 ± 0.4** | **0.004**[**] |
| **Wintering group: Tropical Africa** | | | | |
| *Ficedula albicollis* | -0.9 ± 0.4 | 0.053 | 0.0 ± 0.4 | 0.918 |
| *Ficedula hypoleuca* | -0.4 ± 0.2 | 0.136 | 0.2 ± 0.2 | 0.484 |
| *Hirundo rustica* | -0.2 ± 0.6 | 0.779 | -0.2 ± 0.6 | 0.710 |
| *Merops apiaster* | -0.3 ± 0.4 | 0.396 | 0.1 ± 0.3 | 0.861 |
| *Oriolus oriolus* | -0.2 ± 0.3 | 0.463 | 0.3 ± 0.4 | 0.382 |
| *Saxicola rubetra* | -0.1 ± 0.2 | 0.575 | 0.3 ± 0.2 | 0.280 |

Negative values indicate an advanced passage. Significant slopes are represented in bold typeface.

[**] = $p < 0.01$

[*] = $0.01 < p < 0.05$.

to a lesser extent, in those wintering in the Sahel zone. The peak of passage did not change markedly in species spending the winter in tropical Africa. While there was evidence for an advancement of the beginning of the migration period, the end did not change substantially. This results in a substantial though non-significant increase of the width of the migration window. For example, if we consider the Sahel group in our study, the average migration window changed from ~25 to ~40 days (Fig 1).

The pronounced advance in the peak date of passage in species wintering in North Africa confirms earlier findings that suggested that species not crossing the Sahara are able to better track changing conditions in the breeding grounds [64]. Given the phenological shift in plant productivity and the related change in peak prey abundance for insectivores [65], this result was not surprising. Improved conditions in the Sahel might be responsible for the advancement of migration dates also in species spending the winter in that area. Moreau [66] observed that migratory birds arrive to the Sahel zone at the beginning of the dry season, and throughout the winter they face deteriorating conditions that reach their negative peak when birds are preparing for spring migration. Recent winter rains may have relaxed this situation and allowed for richer foraging conditions during this critical time. Earlier departure with increasing winter rains in the Sahel has been shown for several species in past studies [67]. Interestingly, peak date of passage did not change over the period of the study in species wintering in the forested areas of tropical Africa. These are also the species that start their migration last since migration date is correlated with wintering latitude [68]. In general, the recently described re-greening of the Sahel zone [52, 69] might favour species that extensively use this region as a wintering area or as a stopover site during migration, by allowing faster refueling rates and thus earlier departure [70, 71]. Overall, environmental conditions in the wintering areas are the most likely factor determining regional differences in phenological adjustments. Future studies should address climatic changes in different regions within the wintering range of Eurasian-African migratory species to better understand these patterns.

The earlier start of the migration passage in Ponza is likely due to an earlier departure of the first migrants within each species [10]. The first birds to leave the wintering grounds are usually males [72, 73] and birds belonging to more temperate populations [74]. Our data suggest that both mechanisms play a role. In species wintering in the Sahel, both males and females advanced their peak date of passage, while in species wintering in tropical Africa, this only happened in males. While an increase in protandry seems to explain the earlier passage of tropical winterers, this does not seem to be the explanation for Sahel species. In the latter species, early departing individuals might have advanced their passage while late departing individuals, which possibly originated from more Northern populations [74], did not. There is evidence that individuals do not vary their migration timing over the years [34, 75–80], though this is not true for all species, especially in the case where individuals are able to track environmental cues to adjust their departure [81–84]. The role of phenotypic plasticity in individual departure as opposed to population-specific selection on early departing individuals needs to be further studied for each individual species.

Another explanation for the increased gap between first and last passage migrants is faster migration of the first migrants and/or slower migration of the last ones. There is high variability in the geographical patterns of migration within species and individuals [76, 80], and birds might undertake detours to track favourable habitats along the route [85]. van Noordwijk [86] suggested that faster migration could be achieved by skipping stopovers along the route. Deteriorating conditions in the Sahara Desert or in Northern Africa may cause less efficient stopover and thus an earlier departure, leading to an earlier arrival on Ponza. A skew of the passage phenology towards early migrants would also occur if conditions in Africa affected early migrants differently than late migrants. Northern Wheatears on the neighboring island of Ventotene show better body condition late in the season [74]. This indicates that late migrants might encounter more favourable conditions for refueling in Northern Africa than early migrants.

We do not know whether the change in passage dates on Ponza directly reflects a change in arrival on the breeding grounds. Most birds do not spend more than one day on Ponza before resuming migration [87], thus if they were to delay arrival to their territories, they would have to extend their stopover later on the continent. However, the strong carry-over effects of migration phenology on breeding events [88] indicate that differences in timing observed on Ponza should indeed reflect, at least to a certain extent, the differences in arrival to the breeding grounds. There is some evidence that the change in date of arrival at the breeding grounds in Europe is less steep than the change of passage in the Mediterranean [10, 32, 89], indicating that birds might slow down the pace of their migration when approaching the breeding grounds to fine-tune their arrival. Laying dates have advanced in relation to temperature changes at the breeding grounds [90, 91], to a smaller extent in long-distance compared to short-distance migratory species [92]. The passage data from Ponza confirm this observation, indicating that adjustments to the changing climate might be less pronounced in species wintering the furthest away from their breeding grounds.

Trans-Saharan migratory species show decreasing population trends in many of their breeding grounds in Europe [93, 94]. One of the causes of this decline is the phenological mismatch between the availability of prey at the breeding grounds and the arrival and consequent start of breeding of the birds [95]. The available data do not allow us to draw conclusions about the causes of the intraspecific differences and the mechanisms involved, but they are helpful for developing hypotheses and design future studies. The results of our study call for an intensification of data collection in the form of year-round tracking and long-term data sets at a large geographical scale. More data about the ecology of species, especially in the wintering

quarters, are required to understand the selective pressure acting on migration timing, and to predict future changes and how these will affect population processes.

## Supporting information

**S1 Fig. Frequency distribution of captures by date (Julian day: 1 January = day 1) for 30 species on the island of Ponza between 2002 and 2019.** Average CPUE per day are represented by the black lines, while the moving average is represented by the blue curve. This figure only illustrates general patterns. Note, however, that peak, start, and end of the main migration period were calculated for every year separately for the analysis of timing patterns. (TIF)

**S2 Fig. Yearly dates of passage of 30 species on the island of Ponza between 2002 and 2019.** The blue lines represent the regression line of the start and end of the main migration period, while the black line represents the regression line for peak passage. Black dots represent yearly peak passage dates, while the whiskers represent start and end of the main migration period for every year of the study. (TIF)

**S1 Table. Start and end date of capture operations on Ponza during the 18 years of the study.** (DOCX)

**S2 Table. p-values of the linear regressions of passage date and year for 30 species migrating through Ponza between 2002 and 2019.** Slopes and SE are shown in the main text in Table 2. (DOCX)

## Acknowledgments

We are grateful to all ringers and volunteers helping through the years at the station in Ponza. We are also grateful to the Italian Presidenza della Repubblica for providing qualified personnel from the Castelporziano estate. We also thank Alessandro Orio and Valerio Bartolino for their precious help with the analyses. Two anonymous reviewers greatly contributed to improve the manuscript. This is paper no. 70 from the Piccole Isole Project of the Italian National Institute for Environmental Protection and Research (ISPRA).

## Author Contributions

**Conceptualization:** Massimiliano Cardinale, Fernando Spina, Leonida Fusani.

**Data curation:** Massimiliano Cardinale.

**Formal analysis:** Massimiliano Cardinale, Jonas Hentati Sundberg.

**Funding acquisition:** Leonida Fusani.

**Investigation:** Massimiliano Cardinale.

**Methodology:** Massimiliano Cardinale, Fernando Spina.

**Project administration:** Massimiliano Cardinale.

**Resources:** Leonida Fusani.

**Writing – original draft:** Ivan Maggini, Massimiliano Cardinale.

**Writing – review & editing:** Ivan Maggini, Massimiliano Cardinale, Jonas Hentati Sundberg, Fernando Spina, Leonida Fusani.

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
