## [Decision Letter · Decision Letter 0]

25 May 2020

PONE-D-20-09340

Recent phenology shifts of migratory birds at a Mediterranean spring stopover site: birds wintering in the Sahel advance passage more than tropical winterers

PLOS ONE

Dear Dr. Maggini,

Thank you for submitting your manuscript to PLOS ONE. After careful consideration, we feel that it has merit but does not fully meet PLOS ONE’s publication criteria as it currently stands. Therefore, we invite you to submit a revised version of the manuscript that addresses the points raised during the review process.

The Reviewers made a great work and provided detailed and useful comments. Please consider all them because the suggested changes will add to the quality of the paper. In particular, I strongly advise you to consider with particular attention the comments regarding:

Reviewer #1

A formal comparison between different categories (long vs. short distance migrators, males vs. females)Full reporting of all effects in statistical models, including the non-significant onesEmphasizing the relevance of documenting an advancement in spring migration also in very recent yearsAdding reference to recent literature (please note that the Reviewer very kindly provided a list of potentially relevant and recent papers)

Reviewer #2

Defend or discuss the statistical approach used to estimate phenologyAdd details on the extent to which fieldwork could be performed according to the expected protocol

The major comment by Reviewer #2 deserves also consideration. Adding information on conditions at wintering sites or on passage areas across the Mediterranean would clearly add a lot of interest to the work.

We look forward to receiving your revised manuscript.

Kind regards,

Roberto Ambrosini, Ph.D.

Academic Editor

PLOS ONE

Minor comments:

In Table 1 *Oenante oenante* is written with the species name capitalized. Please correct.

In line 126 and throughout the manuscript please replace “to anticipate” with “to advance”.

Journal Requirements:

Reviewers' comments:

Reviewer's Responses to Questions

**Comments to the Author**

1. Is the manuscript technically sound, and do the data support the conclusions?

Reviewer #1: Partly

Reviewer #2: Yes

2. Has the statistical analysis been performed appropriately and rigorously? 

Reviewer #1: No

Reviewer #2: N/A

3. Have the authors made all data underlying the findings in their manuscript fully available?

Reviewer #1: No

Reviewer #2: No

4. Is the manuscript presented in an intelligible fashion and written in standard English?

Reviewer #1: Yes

Reviewer #2: Yes

5. Review Comments to the Author

Reviewer #1: The study entitled “Recent phenology shifts of migratory birds at a Mediterranean spring stopover site: birds wintering in the Sahel advance passage more than tropical winterers” reports the description of the temporal variation in the dates of passage during the pre-nuptial migration of 18 bird species in a small island of the Mediterranean sea (Ponza). The main result is that most of the species under investigation advanced their passage time, and therefore add to the hundreds of studies reporting similar findings elsewhere in the globe. It also shows that species that migrate longer distances show a smaller temporal variation in phenology (e.g. the slope of the passage date against year) than those migrating further, as well as that males generally show steeper trends than females. The study is merely descriptive but worthy for a least two reasons: 1) it is conducted at an important passage site of which no information on migration time was available to date, and 2) because it was conducted in a rather recent period (2002-2019), and it therefore shows that birds are still shifting their phenology (and I think that this important result is not emphasized enough in the current version of the paper; please see below). I am therefore generally positive about the possibility that the paper would be accepted for publication. However, I am some issues that I would like to raise below in order to improve, I hope, the quality and the readability of the manuscript. I think that they should be easily addressed by the authors.

1) Statistical analyses and implications of the results. The conclusions are reasonable and compatible with the wide literature about the topic. However, I am not sure that the authors could claim that their study shows difference in the phenological variation between the species wintering in Northern Africa and in the Sahel zone and those wintering further South in the forests of central Africa without properly testing for it. I therefore encourage them adopt a proper statistical test in order to check for differences among groups of birds showing different migration strategies (and/or different wintering regions). This is also the case for the supposed differences between the phenology of males and females of some species. I am aware that the number of the species included in the analyses in not very large, but I think that there is the scope for splitting the species at least into two categories. However, if the authors think that this is not possible, in my opinion they should at least justify why they cannot analyse the data and add a caveat in the Discussion by acknowledging the reader about the limitations of their results.

2) Main migration period. The study is focused on the start, end, duration and peak of the so-called ‘main migration period’. The procedure used to define this period seems accurate to me, and it is not my intention to question it. However, since the vast majority of the hundreds of studies on migration phenology published to date usually focus on mean/median and/or first (as well as last) passage dates, I am wondering why the authors opted for this different procedure, which make their results difficult to be compared with the previous ones. I would suggest to at least better justify this decision and to provide some solid reasons why this procedure is better, at least for the case under study, than that used in most of literature.

3) Tables. I have also a concern about how tables have been completed. Frankly, I do not agree with reporting only significant relationships in the tables. I think it is important to report all the associations, including non-significant ones. This good practice is important, for example, to limit publication bias in meta-analytic studies. I therefore strongly encourage the authors to fully fill the tables. In addition, since the main message of the paper is a link between migration strategy and phenological change, I also think that it would be very helpful for the readers (especially for those non-expert in birds) to have some additional information about the species in the first part of the manuscript (other than in the Discussion where they are now reported). A possibility is to add a column in Table 1 reporting the migratory behaviour and/or the main wintering region of each species.

4) Introduction. On the whole, it is concise and well written. However, I was rather surprised not to read any clear mention about the effect of migratory behaviour on variation in migration phenology, also considering that the authors explicitly referred to short- and long-distance migrants in the Discussion. I understand that the authors mentioned where different species spend the wintering period, but information might not be enough for a generalist reader. According to literature, change in phenology depending on migratory behaviour is one of the most consolidated knowledge on the effect of climate change on birds. I would therefore encourage the authors to add a short paragraph on this (including relevant references), linked to that one focused on the effects of NAOI and wintering areas.

5) Discussion. As briefly mentioned above, I think that one the main interesting point of the study is that migration data have been collected rather recently. I think that the paper would benefit from the inclusion of an additional paragraph where the authors emphasize that the phenological shifting is still ongoing and compare their recent trends with those obtained by previous studies performed on very long time-series (e.g. Askeyev et al. 2009. Climate Research, 38: 189-192; Kolářová et al. 2015. Climate Research, 63: 91-98).

6) Literature cited. References quoted in the paper are generally relevant. However, I noted that only a few very recent papers (after 2015) have been cited, despite the vast literature produced in the last years. This is especially the case for Introduction, in which some very relevant and recent papers are missing. I therefore suggest to add some of them in order to better introduce the concepts, which will be developed through the manuscript. Please, find below some missing articles which are very relevant to the purpose of the study, and deserve to be cited (in chronological order).

- Kluen, E., Nousiainen, R., & Lehikoinen, A. (2017). Breeding phenological response to spring weather conditions in common Finnish birds: resident species respond stronger than migratory species. Journal of Avian Biology, 48(5), 611-619.

- Samplonius, J. M., Bartošová, L., Burgess, M. D., Bushuev, A. V., Eeva, T., Ivankina, E. V., ... & Mänd, R. (2018). Phenological sensitivity to climate change is higher in resident than in migrant bird populations among European cavity breeders. Global change biology, 24(8), 3780-3790.

- Ambrosini, R., Romano, A., & Saino, N. (2019). Changes in migration, carry-over effects, and migratory connectivity. Effects of Climate Change on Birds, 93.

- Radchuk, V., Reed, T., Teplitsky, C., Van De Pol, M., Charmantier, A., Hassall, C., ... & Avilés, J. M. (2019). Adaptive responses of animals to climate change are most likely insufficient. Nature communications, 10(1), 1-14.

- Horton, K. G., La Sorte, F. A., Sheldon, D., Lin, T. Y., Winner, K., Bernstein, G., ... & Farnsworth, A. (2020). Phenology of nocturnal avian migration has shifted at the continental scale. Nature Climate Change, 10(1), 63-68.

Minor comments:

LL 62-65. Maybe add some references here.

L 74. “near-passerine”. I think this should be defined, especially in a very generalist journal like the present one.

LL 89-92 and Table 1. It is ok to include information about the total number of individuals captured per species. However, because the authors declared that they selected the species to be included in the analyses based on the yearly number of captures, I think that an important missing information is also the range of individuals captured per species per year. In addition, it is not clearly described which criterion was used to decide whether including or not a species. For example, what is the minimum number of specimens per year? In addition, did the authors exclude some years within species because they could not collect information on a minimum number of individuals? In my opinion, all this information should be added to improve the understanding of the procedures used.

LL 113-115. Interesting that all the species show a significant temporal variation (advance or delay) in the phenology of migration. This is a quite unexpected result, if we compare it with the available literature. How do the authors explain this result? Is possible that the procedure used to identify the main migration period has somehow affected this result?

L 244. Publication number XX. Maybe there is an error here.

Reviewer #2: This paper aims to analyse the spring migratory phenology of 30 species of birds in a Mediterranean island, for a period of 18 years. Authors found a broadening of the time of passage overall, due to an advancement of the passage by mostly short-distance migrants.

The paper is well written, it is easy to follow and the concepts and literature used are within the expected current state of the art. A have, however, some comments to do:

Main comments:

The data set used in the study is impressive, and this is a robustness of the paper. However, the paper is totally descriptive, and the authors ‘only’ analyse a linear effect of year on the object, dependent variables (annual values of the start, peak and end of migration period). It would be really nice (though I acknowledge that this would entail much more work) if the authors may also look for the potential effects of conditions at wintering sites or during the passage across the Mediterranean. These analyses would allow a richer discussion around the observed changes/results. This is what I would also expect in a journal like PLOS ONE.

Minor comments:

L94-96. Quite often, mist nets must be closed due to very adverse weather (heavy rain, too strong winds) or even logistical reasons (lack of personal in given moments). Even though the authors say that the fieldwork was done on a daily basis, I would like to know to what extent this protocol was done with a 100% of, most likely, whether there were some gaps. In these cases, many authors think that replacing raw data by theoretical distribution regressions is better, since these last allow to: deal with gaps in the data set and smooth the potential effects of the variables affecting capture rates (e.g., birds are more likely to be capture in days without wind, or in cloudy mornings as compared to very sunny days). Furthermore, these theoretical distributions would also allow dealing with years when the campaign started after the onset of migration, or ended before its end. Authors should discuss/defend their statistical approach and explain why not dealing with the use of theoretical distribution curves.

L103. A good reason to use theoretical curves?

L106-109. For me, this would be an insufficient explanation. There can be other distribution curves that might fit reasonably well to your data.

L139. Please, this should be mention in Methods section.

L157. Overall, the discussion might be benefited if the authors would also consider to test for the effects of conditions in the wintering/passing areas on their dependent variables.

Table 1. Please, add SE to the beta parameter estimtes (slope).

Table 2. Too many NAs in the start and end of migration. Why? Maybe this might be improved by using theoretical curves? If not, please limit the analyses to the peak passage parameter.

6. PLOS authors have the option to publish the peer review history of their article (what does this mean?). If published, this will include your full peer review and any attached files.

Reviewer #1: No

Reviewer #2: No

---

## [Author Response · Author response to Decision Letter 0]

8 Jul 2020

Please find our answers to the reviewers' and editor's comments in the attached file.

---

## [Decision Letter · Decision Letter 1]

11 Aug 2020

PONE-D-20-09340R1

Recent phenological shifts of migratory birds at a Mediterranean spring stopover site: species wintering in the Sahel advance passage more than tropical winterers

PLOS ONE

Dear Dr. Maggini,

Thank you for submitting your manuscript to PLOS ONE. After careful consideration, we feel that it has merit but does not fully meet PLOS ONE’s publication criteria as it currently stands. Therefore, we invite you to submit a revised version of the manuscript that addresses the points raised during the review process.

This new version was revised by the Reviewer that raised the most serius concerns on the first version. He/she was rather satisfied by this new version, but he/she also asked for some further minor changes. We think you can easily modify the manuscript to acccount for these further suggestions. Please submit your revised manuscript by Sep 25 2020 11:59PM. If you will need more time than this to complete your revisions, please reply to this message or contact the journal office at plosone@plos.org. Please include the following items when submitting your revised manuscript:

We look forward to receiving your revised manuscript.

Kind regards,

Roberto Ambrosini, Ph.D.

Academic Editor

PLOS ONE

Reviewers' comments:

Reviewer's Responses to Questions

**Comments to the Author**

1. If the authors have adequately addressed your comments raised in a previous round of review and you feel that this manuscript is now acceptable for publication, you may indicate that here to bypass the “Comments to the Author” section, enter your conflict of interest statement in the “Confidential to Editor” section, and submit your "Accept" recommendation.

Reviewer #1: All comments have been addressed

2. Is the manuscript technically sound, and do the data support the conclusions?

Reviewer #1: Yes

3. Has the statistical analysis been performed appropriately and rigorously? 

Reviewer #1: Yes

4. Have the authors made all data underlying the findings in their manuscript fully available?

Reviewer #1: Yes

5. Is the manuscript presented in an intelligible fashion and written in standard English?

Reviewer #1: Yes

6. Review Comments to the Author

Reviewer #1: I am rather satisfied by the revision made by the authors because they provided all the main suggested changes to the manuscript. In my opinion, the paper has considerably improved and I therefore suggest it for publication. I have only a few additional minor comments (lines numbers refer to the version including track changes):

LL 151-152. Please add references used to define wintering areas and to associate species to them.

L 211.5 rather than 22…a huge difference!

LL 211-237. The reader is lost until the last line of this long paragraph because the table and the figure are cited only at end. I suggest to mention them before.

L 237. Fig. S2, not S2 Fig.

Table 2. I am not sure that P values between 0.05 and 0.10 should be mentioned in a table, even considering that exact P-values are provided in the supplementary materials.

LL 383-384. There is some evidence that in many species the advance in arrival date in the breeding area is less steeper than the passage date in the Mediterranean area. This point might deserve a short discussion and some references (please see below).

Both, C. (2010). Flexibility of timing of avian migration to climate change masked by environmental constraints en route. Current Biology, 20(3), 243-248.

Jonzén, N., Lind´en, A., Ergon, T., Knudsen, E., Vik, J.O., Rubolini, D., Piacentini, D., Brinch, C., Spina, F., Karlsson, L., Stervander, M., Andersson, A., Waldenstr¨om, J., Lehikoinen, A., Edvardsen, E., Solvang, R. & Stenseth, N. C. (2006). Rapid advance of spring arrival dates in long-distance migratory birds. Science 312, 1959–1961.

Bitterlin, L. R., & Van Buskirk, J. (2014). Ecological and life history correlates of changes in avian migration timing in response to climate change. Climate Research, 61(2), 109-121.

LL 390-391. A very important study to be cited here is: Dunn, P. O., & Møller, A. P. (2014). Changes in breeding phenology and population size of birds. Journal of Animal Ecology, 729-739.

7. PLOS authors have the option to publish the peer review history of their article (what does this mean?). If published, this will include your full peer review and any attached files.

Reviewer #1: No

---

## [Author Response · Author response to Decision Letter 1]

12 Aug 2020

Reviewer #1: I am rather satisfied by the revision made by the authors because they provided all the main suggested changes to the manuscript. In my opinion, the paper has considerably improved and I therefore suggest it for publication. 

We thank the reviewer for the renewed assessment of our manuscript, and especially for her/his comments on the previous version that helped us to largely improve this new version.

I have only a few additional minor comments (lines numbers refer to the version including track changes):

LL 151-152. Please add references used to define wintering areas and to associate species to them.

The references are given in line 95 (new line numbering)

L 211.5 rather than 22…a huge difference!

We agree. This was due to a mistake in the previous analysis that we corrected in the new version. In the previous version, the large number of significant slopes caught the attention of the reviewers and was highly unlikely.

LL 211-237. The reader is lost until the last line of this long paragraph because the table and the figure are cited only at end. I suggest to mention them before.

We moved the reference to table and figure at the beginning of the paragraph.

L 237. Fig. S2, not S2 Fig.

This was suggested in the information for authors. We will change this if requested.

Table 2. I am not sure that P values between 0.05 and 0.10 should be mentioned in a table, even considering that exact P-values are provided in the supplementary materials.

We removed the highlights from slopes with p between 0.05 and 0.10 (we did the same in Table 3).

LL 383-384. There is some evidence that in many species the advance in arrival date in the breeding area is less steeper than the passage date in the Mediterranean area. This point might deserve a short discussion and some references (please see below).

Both, C. (2010). Flexibility of timing of avian migration to climate change masked by environmental constraints en route. Current Biology, 20(3), 243-248.

Jonzén, N., Lind´en, A., Ergon, T., Knudsen, E., Vik, J.O., Rubolini, D., Piacentini, D., Brinch, C., Spina, F., Karlsson, L., Stervander, M., Andersson, A., Waldenstr¨om, J., Lehikoinen, A., Edvardsen, E., Solvang, R. & Stenseth, N. C. (2006). Rapid advance of spring arrival dates in long-distance migratory birds. Science 312, 1959–1961.

Bitterlin, L. R., & Van Buskirk, J. (2014). Ecological and life history correlates of changes in avian migration timing in response to climate change. Climate Research, 61(2), 109-121.

We added a sentence to acknowledge this (l. 302-304).

LL 390-391. A very important study to be cited here is: Dunn, P. O., & Møller, A. P. (2014). Changes in breeding phenology and population size of birds. Journal of Animal Ecology, 729-739.

We added this reference.

See also the attached file.

---

## [Editor Report · Decision Letter 2]

8 Sep 2020

Recent phenological shifts of migratory birds at a Mediterranean spring stopover site: species wintering in the Sahel advance passage more than tropical winterers

PONE-D-20-09340R2

Dear Dr. Maggini,

We’re pleased to inform you that your manuscript has been judged scientifically suitable for publication and will be formally accepted for publication once it meets all outstanding technical requirements.

Kind regards,

Roberto Ambrosini, Ph.D.

Academic Editor

PLOS ONE
---

## [Editor Report · Acceptance letter]

10 Sep 2020

PONE-D-20-09340R2 

Recent phenological shifts of migratory birds at a Mediterranean spring stopover site: species wintering in the Sahel advance passage more than tropical winterers 

Dear Dr. Maggini:

I'm pleased to inform you that your manuscript has been deemed suitable for publication in PLOS ONE. Congratulations! Your manuscript is now with our production department. 

Kind regards, 

on behalf of

Dr. Roberto Ambrosini 

Academic Editor

PLOS ONE